# Exportin 1 as a Therapeutic Target to Overcome Drug Resistance in Lung Cancer

**DOI:** 10.3390/cells14241991

**Published:** 2025-12-15

**Authors:** Maria Vittoria Di Marco, Alessandro Gasparetto, Roberto Chiarle, Claudia Voena

**Affiliations:** 1Department of Molecular Biotechnology and Health Sciences, University of Turin, 10124 Turin, Italy; mariavittoria.dimarco@unito.it (M.V.D.M.); alessandro.gasparetto@unito.it (A.G.); 2Department of Pathology, Boston Children’s Hospital, Boston, MA 02115, USA; 3Department of Hemopathology, European Institute of Oncology (IEO), 20141 Milan, Italy

**Keywords:** NSCLC, XPO1, drug resistance, TP53

## Abstract

Non-small cell lung cancer (NSCLC) remains the leading cause of cancer-related mortality, with therapeutic resistance continuing to limit long-term responses. Among emerging resistance mechanisms, dysregulation of nucleocytoplasmic transport has gained attention for its ability to inactivate tumor suppressor pathways. Exportin 1 (XPO1), the primary nuclear export protein, is frequently overexpressed in NSCLC and promotes the cytoplasmic mislocalization of proteins involved in cell cycle control, apoptosis, and DNA repair. This includes key regulators such as p53, FOXO, and RB, whose inactivation supports tumor progression and therapy resistance. Inhibition of XPO1 with selective inhibitors of nuclear export (SINE) compounds, including selinexor, has demonstrated the ability to restore nuclear localization and function of these proteins, thereby enhancing cellular sensitivity to DNA-damaging agents, kinase inhibitors, and immunotherapies. In preclinical NSCLC models, XPO1 inhibition has shown efficacy both as monotherapy and in combination strategies, with particular promise in KRAS- and EGFR-driven tumors. This review explores the role of XPO1 in NSCLC biology and therapy resistance, the rationale for targeting nuclear export, and the current landscape of XPO1-directed clinical development in lung cancer.

## 1. Introduction

Lung cancer remains the leading cause of cancer-related mortality worldwide, accounting for over 1.7 million deaths annually, despite advances in early detection and therapy [1]. Approximately 80% of cases are attributable to active tobacco smoking, although a significant fraction arises in individuals with minimal or no smoking history, often driven by distinct genetic and environmental factors [2]. Histologically, lung cancer is classified into Small-Cell Lung Cancer (SCLC), which is highly aggressive and strongly linked to tobacco exposure, and Non-Small Cell Lung Cancer (NSCLC), which comprises approximately 85% of cases and exhibits slower progression but is frequently diagnosed at advanced stages. NSCLC is a molecularly heterogeneous disease, and the most common molecular drivers include activating mutations in *EGFR*, *KRAS*, and *BRAF*, as well as gene rearrangements involving *ALK*, *ROS1*, and *RET*, which promote uncontrolled cell growth and survival. Additional alterations in *MET*, *HER2*, and *NTRK* further define distinct molecular subsets. Collectively, the heterogeneity of lung cancer in terms of histology, molecular drivers, and clinical behavior underscores the necessity for targeted therapeutic strategies and accentuates the importance of emerging molecular targets in overcoming or potentially preventing drug resistance [3,4].

Intracellular compartmentalization and trafficking of molecules play a critical role in complex and essential cellular processes. Aberrant nucleocytoplasmic transport of tumor suppressor proteins and cell cycle regulators, mediated by importins and exportins, can result in tumorigenesis and inactivation of apoptosis. Several malignancies, including lung cancer, overexpress these nuclear transport receptors as an oncogenic feature. Pharmacologic targeting of nuclear export has demonstrated antitumor efficacy [5]. In this review article, we provide an overview of the mechanisms of nuclear export mediated by the most relevant carrier protein, exportin 1 (XPO1), and how its inhibition is a potential approach to prevent and treat acquired drug resistance.

## 2. XPO1 Role in Physiology and Cancer

The nuclear envelope, composed of inner and outer membranes, functions as a selective barrier that prevents passive diffusion of molecules larger than approximately 40 kDa between the nucleus and cytoplasm [6]. To maintain cellular homeostasis, eukaryotic cells depend on active nucleocytoplasmic transport, which is vital for gene regulation and signal transduction [7]. This transport occurs via nuclear pore complexes (NPCs), channels within the nuclear envelope that facilitate bidirectional movement of molecules [8]. Transport receptors from the karyopherin-β family, including importins and exportins, manage the specific transfer of cargo across the NPC [9].

Exported proteins contain leucine-rich motifs known as nuclear export signals (NES), which are specifically recognized by exportins [10]. Export is driven by RanGTP, which forms a complex with exportin and cargo inside the nucleus. This complex moves through the NPC, and after RanGTP hydrolysis in the cytoplasm, it releases the cargo and recycles the exportin back to the nucleus [5].

Among the seven exportins identified in eukaryotes, XPO1 (also known as CRM1) is the most widespread and essential. It transports over 200 substrates, including key tumor suppressors like p53, RB, and p27 [11]. Its role extends beyond protein shuttling; XPO1 is responsible for exporting several different RNA types, including rRNAs, snRNAs, mRNAs, microRNAs, and tRNAs [12].

Due to its role in regulating oncogenes and tumor suppressors, XPO1 has been extensively studied in cancer research. Overexpression of XPO1 is common across multiple malignancies, including pancreatic, lung, gastric, prostate, and colorectal cancers, and frequently correlates with aggressive disease and poor prognosis [5,13]. In NSCLC, aberrant XPO1 expression has been linked to disease progression, radiation resistance, and decreased overall survival [14]. Beyond transcriptional upregulation, *XPO1* is subject to additional genetic alterations, including missense hotspot mutations and copy number amplifications. A recurrent E571K point mutation in *XPO1* has been documented most prominently in B-cell malignancies, where it enhances binding affinity to nuclear export signals and alters nucleocytoplasmic transport [15,16]. Although rare, *XPO1* mutations have also been observed in solid tumors, including NSCLC: in a pan-cancer survey, approximately 2.8% of NSCLC cases exhibited *XPO1* alterations (copy number cthanges or mutations) [17], and in a large NSCLC cohort, 26 tumors carried *XPO1* mutations and 24 showed amplifications [18]. These copy number gains often involve moderate amplifications [19]. XPO1 overexpression is also associated with drug resistance on account of the export of drug targets such as topoisomerase II and galectin-3 [5].

## 3. XPO1 Pharmacological Inhibition

The oncogenic significance of XPO1 has led to efforts to target it therapeutically. Cancer cells, with their high proliferative and metabolic requirements, are particularly vulnerable to disruptions in nuclear export. Therefore, inhibiting nuclear export, either alone or in combination with conventional therapies, has become a promising treatment approach [5].

While inhibitors of nuclear import, such as importazole, INI-43, and ivermectin, are still in preclinical development [5], the progress in developing exportin inhibitors has been rapid. Early efforts utilized natural products like leptomycin B [20,21], anguinomycins [20], and ratjadones [22].

Leptomycin B (LMB) binds covalently and irreversibly to Cys528 of XPO1, blocking its ability to interact with cargo. Despite its specificity, clinical trials were discontinued due to severe systemic toxicities, even at low doses [23,24]. Structural analogs, such as leptomycin A, anguinomycins, and ratjadones, have shown limited therapeutic potential [5]. Conversely, felezonexor (SL-801) binds reversibly to Cys528 and also promotes XPO1 degradation through neddylation, potentially improving tolerability. A Phase I trial (NCT02667873) with SL-801 was recently concluded; interim results showed that the drug was well tolerated and 29% of patients achieved stable disease [25].

These advancements led to the development of the Selective Inhibitors of Nuclear Export (SINE) class of compounds, including KPT-185, KPT-251, KPT-276, selinexor, eltanexor, and verdinexor. These orally administered compounds bind to Cys528 on XPO1 in a slow, reversible manner, offering a better safety profile [5,26,27]. In addition, unlike LMB, many SINEs not only inhibit XPO1 function but also promote its proteasomal degradation, leading to a progressive decrease in protein levels. This reduction results from proteasome-dependent degradation rather than reduced transcription, indicating a dual mechanism whereby SINEs exert both covalent inhibition of export activity and active elimination of the export receptor itself [28]. Globally, selinexor has been administered to over 2100 patients. Common adverse events include low-grade nausea (62%), fatigue (60%), anorexia (51%), thrombocytopenia (42%), and vomiting (37%), which have generally been readily managed with standard supportive care measures [29].

Selinexor has received FDA approval for the treatment of multiple cancer types. In July 2019, it was granted accelerated approval for the treatment of relapsed or refractory multiple myeloma in combination with dexamethasone [30]. In June 2020, FDA approval was expanded to include relapsed/refractory diffuse large B-cell lymphoma (DLBCL), based on the SADAL trial results, which showed an overall response rate (ORR) of approximately 29% with durable responses in some patients [31].

Selinexor is currently under investigation in several solid tumor settings. A Phase III trial is evaluating selinexor as maintenance therapy in *TP53* wild-type endometrial cancer [32]. In ovarian and breast cancers, selinexor is being tested in combination with carboplatin, paclitaxel, eribulin, or topotecan [33]. In NSCLC, a Phase I/II study is evaluating selinexor in combination with docetaxel in patients with previously treated KRAS-mutant disease [34]. Pediatric trials are also ongoing, including a Phase II study in relapsed or refractory Wilms’ tumor, rhabdoid tumors, and other pediatric solid malignancies (NCT05985161).

## 4. XPO1 Function and Mechanism of Inhibition

Due to its role in various malignancies, XPO1 has become a promising therapeutic target to prevent tumor progression and relapse, and SINE compounds have been developed. Understanding the mechanism of action of XPO1 inhibitors remains an ongoing process that is crucial for designing successful preclinical and clinical studies (Figure 1).

### 4.1. Nuclear Trafficking

XPO1 is involved in the export of nearly 220 proteins bearing NESs. These substrates are critical for tumor cells, as XPO1 cargos are involved in the majority of hallmarks of cancer: sustained proliferation (c-ABL, c-Myc, c-Met, EGFR), evading growth suppressors (p21, p27), genome instability (p53, DNA topoisomerases), resisting cell death (survivin, Bok, FOXO), enabling replicative immortality (TERT), inducing angiogenesis (Fbw7, COMMD1), activating invasion and metastasis (FBXL5, snail, APC, c-Pml), deregulating cellular energetics (eIF4E), and tumor-promoting inflammation (Cox-2) [5,11,35]. Remarkable is the role of XPO1 in exporting the BCL-ABL fusion protein in Chronic Myeloid Leukemia. BCR–ABL chimeric protein, which constantly stimulates the proliferation of myeloid cells, is exported to the cytoplasm of cancer cells, where it activates the PI3K/Akt pathway. XPO1 inhibition traps BCR–ABL in the nucleus, re-sensitizing leukemia cells to the BCR–ABL inhibitor imatinib and resulting in a substantial reduction in tumor cell proliferative potential with limited toxicity to normal myeloid precursors [36]. While early models suggested that XPO1 inhibitors act primarily by restoring nuclear localization of tumor suppressors, experimental evidence shows that these agents exert antitumor activity even in cells lacking functional p53, RB, or p21, indicating a broader mechanism beyond tumor suppressor retention [26,35].

### 4.2. RNA Export

XPO1 plays a pivotal role in RNA metabolism by exporting multiple classes of RNAs. Ribosomal subunits (40S and 60S pre-ribosomal particles) rely on XPO1 for translocation from the nucleolus to the cytoplasm, a critical step for ribosome maturation and translation (Thomas, 2003) [37]. Small nuclear RNAs (snRNAs), which are essential for pre-mRNA splicing, form export-competent complexes with PHAX and RanGTP, which are subsequently exported by XPO1 and re-imported as mature snRNPs [38]. Additionally, XPO1 mediates alternative export pathways for microRNAs and tRNAs [12]. Selective export of mRNAs encoding oncoproteins occurs via XPO1 in cooperation with adaptor proteins such as LRPPRC, eIF4E, NXF3, and HuR, linking nuclear export to the post-transcriptional regulation of tumorigenic pathways [39]. Inhibition of XPO1 thus disrupts both protein and RNA trafficking, impairing ribosome biogenesis, mRNA translation, and RNA processing. A preclinical study demonstrated the potential of exploiting RNA trafficking inhibition in diffuse large B-cell lymphoma, where highly aggressive subtypes rely on Hsp90 activation, resulting in increased RNA nuclear shuttling. Inhibition of the eIF4E factor induced tumor control, indicating RNA nuclear export as a druggable target in cancer [39,40,41].

### 4.3. Cell Cycle

XPO1 influences cell cycle progression by controlling the nuclear-cytoplasmic distribution of numerous cyclins and cyclin-dependent kinase (CDK) inhibitors. Inhibition of XPO1 using SINEs, such as selinexor, has been shown to trigger cell cycle arrest at both the G1 and G2/M phases in cancer cells, often independently of key tumor suppressors, including p53, RB, and p21 [26,27]. By broadly affecting the localization of cell cycle regulators, XPO1 inhibitors reduce proliferation and enhance apoptotic susceptibility, thereby contributing to their antitumor activity. Beyond its role in nuclear export, XPO1 also performs critical mitotic functions. It localizes to kinetochores during mitosis and is essential for proper microtubule nucleation and mitotic spindle assembly, underscoring the export-independent roles of XPO1 in cell division. [42,43,44,45,46]

### 4.4. Epigenetics Activity

Emerging evidence indicates that XPO1 modulates epigenetic states by exporting chromatin regulators and transcription factors. Proteins involved in histone modification, DNA repair, and transcriptional repression rely on nuclear-cytoplasmic shuttling via XPO1, suggesting that its inhibition may alter chromatin accessibility and gene expression programs in cancer cells [5]. XPO1 involvement in epigenetic regulation extends beyond its role in nuclear export. Recent studies have highlighted its role in chromatin docking, a process crucial for transcriptional regulation. For instance, a recent study demonstrated that XPO1 serves as an adaptor for the transcription factor-mediated docking of chromatin at the nuclear pore complex, promoting stronger transcription, primarily by enhancing TF binding to DNA [47].

### 4.5. Modulation of the Tumor Microenvironment

It has been reported that XPO1 also influences the tumor microenvironment (TME) by regulating the localization and activity of signaling proteins and cytokine modulators. The nuclear export of transcription factors, such as NF-κB and HIF-1α, coordinates tumor-promoting inflammation, angiogenesis, and immune evasion [35]. Inhibition of XPO1 reduces the nuclear export of these factors, attenuating pro-tumorigenic signaling and enhancing the susceptibility of cancer cells to immune-mediated clearance and chemotherapy [5].

XPO1 also plays a crucial role in modulating the polarization and activity of immune cells within the TME. Overexpression or mutation of XPO1 is associated with immunosuppressive TMEs, characterized by reduced infiltration of cytotoxic CD8+ T cells and natural killer (NK) cells, increased expression of immune checkpoints, and elevated activity of myeloid-derived suppressor cells (MDSCs) [48]. Mechanistically, XPO1 inhibition has been shown to repolarize macrophages from an M2-like pro-tumor phenotype to an M1-like anti-tumor phenotype, reduce inhibitory checkpoint expression on myeloid cells, and convert MDSCs into immunostimulatory phenotypes, collectively enhancing T cell and NK cell cytotoxicity [49,50]. Furthermore, selinexor downregulates HLA-E on malignant cells, relieving NKG2A-mediated inhibitory signaling and boosting NK- and CD8+ T cell-mediated anti-tumor responses [51,52]. Notably, XPO1 inhibition also enhances the efficacy of adoptive cell therapies, as pre-treatment of tumor cells with selinexor sensitizes them to Chimeric Antigen Receptor (CAR)-T cells and bispecific antibody-mediated cytotoxicity, illustrating a dual role in directly modulating cancer cells and reshaping the immune landscape [53,54].

Despite these promising findings, XPO1 can also bind to *NFAT* regulatory regions in T cells, promoting T cell activation. Inhibition of XPO1 has been shown to reduce overall T cell stimulation, highlighting a context-dependent effect on immune responses [55]. Further studies are warranted to fully elucidate the mechanisms by which XPO1 regulates immunity in different cellular and tumor contexts.

Collectively, these data indicate that XPO1 is a critical regulator of the TME, and its pharmacologic inhibition may represent a promising strategy to enhance anti-tumor immunity and improve the efficacy of cancer therapies.

## 5. XPO1 Inhibition in Lung Cancer Drug Resistance

Lung cancer remains the leading cause of cancer-related mortality among both men and women, encompassing both NSCLC and SCLC subtypes [56]. Despite advances in early detection and targeted therapies, long-term survival remains limited, mainly due to the emergence of drug-tolerant persister cells that survive initial treatments and contribute to relapse or metastasis [57]. A growing body of evidence implicates dysregulation of nuclear-cytoplasmic transport in this process, with XPO1 frequently overexpressed in lung cancers [58] (Figure 2). Given the substantial heterogeneity observed across lung cancer subtypes, therapeutic strategies that simultaneously target multiple critical pathways, such as nuclear export and apoptosis evasion, may provide enhanced clinical benefit.

### 5.1. Non-Small Cell Lung Cancer

Histologically, about 85% of lung cancers are classified as NSCLCs [1]. Most of these cases are diagnosed at an advanced stage, preventing the possibility of curative surgery. Targeted therapies have improved survival for patients with oncogene-driven NSCLC; however, patients eventually develop acquired resistance, underscoring an urgent need for new treatments with lower toxicity and better outcomes. Aberrant expression of XPO1 is well recognized in NSCLC and is linked to poor OS. Ectopic expression of XPO1 in the lung epithelial cell line BEAS-2B led to cellular transformation, suggesting that upregulation of XPO1 may be a critical pathway in the malignant transformation of lung epithelial cells [58]. A comprehensive bioinformatics analysis of *XPO1* alterations in NSCLC, including 5792 samples and 5644 patients, revealed that XPO1 copy number alterations were significantly associated with poor OS [19]. While *XPO1* mutations are relatively rare, their presence correlates with poor prognosis and higher tumor mutational burden, highlighting their potential as prognostic biomarkers [19]. Interestingly, no overlap was observed between *XPO1* mutations and amplification in any patient samples, and there were no significant differences in gender or age between *XPO1* statuses [19]. Preclinical studies demonstrate that pharmacologic inhibition of XPO1 with agents such as selinexor suppresses NSCLC growth both in vitro and in xenograft models [13,59].

### 5.2. Platinum Resistance in NSCLC

Platinum-based chemotherapy remains a cornerstone of first-line treatment in advanced NSCLC; however, the development of resistance is virtually inevitable and leads to disease recurrence and poor survival outcomes. One of the primary mechanisms underlying platinum resistance involves the upregulation of DNA damage repair pathways, including key mediators such as CHEK1, MLH1, MSH2, RAD51, and PMS2, which enable tumor cells to withstand platinum-induced cytotoxicity [60,61]. XPO1 overexpression has been associated with platinum resistance and inferior prognosis across solid tumors, including lung cancer, by promoting the cytoplasmic mislocalization of critical tumor suppressors and DNA repair regulators [29]. Preclinical studies have demonstrated that inhibition of XPO1 with SINE compounds suppresses cellular proliferation and restores platinum sensitivity, partly through nuclear retention of ERK1/2, IκBα, NF-κB, and p65, leading to inhibition of NF-κB signaling and enhanced apoptosis [62]. Mechanistically, selinexor reduces Myc binding to the *RAD51* and *CHEK1* promoters, thereby decreasing the expression of DNA repair proteins and sensitizing tumor cells to platinum-induced DNA damage [26,27].

### 5.3. Resistance to Targeted Therapy

Tyrosine kinase inhibitors (TKIs) have been developed for the treatment of NSCLC with actionable oncogene drivers, including *EGFR*, *KRAS* and *ALK*. Even though these drugs exhibit substantial antitumor activity, their clinical efficacy is often limited by the emergence of drug resistance [57].

In EGFR-mutant NSCLC, secondary resistance mechanisms, including the T790M gatekeeper mutation and activation of bypass pathways (e.g., MET, AXL), limit the durability of TKIs. Inhibition of XPO1 restores nuclear localization of tumor suppressors and enhances apoptosis, thereby resensitizing resistant cells to treatment. For example, NSCLC cell lines harboring *EGFR* T790M mutations, which confer resistance to gefitinib and erlotinib, remain sensitive to selinexor [59,63,64].

Beyond T790M and bypass signaling, other mechanisms of acquired resistance to gefitinib involve RNA helicase–mediated regulation of β-catenin. Elevated expression of the DEAD-box helicase DDX17 in NSCLC disrupts E-cadherin/β-catenin complexes, leading to nuclear β-catenin accumulation and transcription of resistance-associated genes such as *AXIN* and *cyclin D1* [65]. Importantly, DDX17 contains both NLS and NES sequences that enable nucleocytoplasmic shuttling via XPO1. Mutation of these sequences reduces gefitinib resistance in PC9 NSCLC cells compared with wild-type DDX17 [65]. These observations suggest that nuclear transport plays a crucial role in modulating TKI sensitivity, indicating that inhibition of XPO1 could represent a promising strategy to block the export of DDX17 and related DEAD-box helicases [29]. Collectively, these data underscore the functional intersection between XPO1 activity and EGFR-driven signaling, supporting the relevance of XPO1 as a modulator of both primary and acquired resistance in EGFR-mutant NSCLC.

KRAS-mutant NSCLC, long considered undruggable, has recently seen therapeutic breakthroughs with KRAS G12C inhibitors. Nonetheless, resistance develops rapidly, often through the activation of survival pathways. XPO1 plays a pivotal role in contributing to therapeutic resistance by maintaining survival pathways that limit the effectiveness of KRAS G12C inhibitors. Inhibition of XPO1 restores sensitivity to KRAS G12C inhibitors and synergizes with Bcl-xL blockade or cisplatin, inducing apoptosis in vitro [14,18,59,66,67]. Patient-derived xenograft (PDX) models of KRAS-, EGFR-, and MET-driven NSCLC also respond to selinexor, underscoring its broad activity [18]. Furthermore, selinexor demonstrated antiproliferative activity against a panel of 11 NSCLC cell lines, showing different genetic landscapes: 5 of the 11 NSCLC cell lines had either a *K-* or *N-RAS* mutation, 1 of the 11 had loss of *PTEN*, and 2 of the 11 NSCLC cell lines had *PIK3CA* mutations. Thus, KPT-330 may have unique abilities to inhibit the growth of tumor cells containing mutant tumor suppressor genes or activated oncogenes. In addition, a Phase I/II trial combining selinexor with docetaxel in pretreated KRAS-mutant NSCLC showed promising efficacy, particularly in *TP53* wild-type tumors, with manageable toxicity [34]. These findings indicate that KRAS-mutant tumors exhibit a functional dependence on XPO1-mediated nuclear export, which sustains survival pathways and limits the efficacy of KRAS-targeted therapies.

Although XPO1 has emerged as a promising therapeutic target for overcoming resistance in EGFR- and RAS-driven NSCLC, experimental evidence in *ALK*-rearranged NSCLC remains limited. Recent preclinical studies indicated that XPO1 inhibition may represent an effective strategy in ALK+ NSCLC. In vitro and in vivo analyses, including patient-derived cells, tumor organoids, and xenograft models, revealed strong synergy between SINE and ALK-TKIs, characterized by G1 cell-cycle arrest, p53 accumulation, and apoptosis. Notably, the combination therapy significantly reduced tumor growth in TKI-resistant models, enhancing antiproliferative and pro-apoptotic effects compared to monotherapy and suggesting its potential to overcome established resistance mechanisms [68]. Taken together, these preliminary data indicate that *ALK*-rearranged tumors may also depend on XPO1-regulated nuclear export pathways, positioning XPO1 inhibition as a potential strategy to enhance or restore ALK-TKI sensitivity.

### 5.4. Neuroendocrine Transformation

Neuroendocrine (NE) transformation indicates progenitor cell plasticity and is linked to resistance to therapeutic agents. In lung adenocarcinomas, the shift to an aggressive neuroendocrine phenotype resembling SCLC is associated with poor clinical outcomes. The development of more potent and specific inhibitors of oncogenic drivers may increase selective pressure and thus the occurrence of such histological transdifferentiation [69]. NE transformation has been reported in approximately 3–10% of EGFR-mutant NSCLCs treated with TKIs [70], and although less common, has also been observed in ALK- [71], ROS1- [72], RET- [73,74], and other genomic contexts, indicating a clinically relevant but infrequent pathway of resistance. Although limited in number, existing reports of patients with squamous transformation have shown the retention of characteristic EGFR mutations, EML4–ALK fusions, or ROS-1 rearrangements from the adenocarcinoma, supporting the idea of lineage transformation [72,75,76,77]. The poor prognosis of patients after NE transformation highlights the need to develop strategies to limit or prevent plasticity and to treat NE transformation more effectively.

One of the hallmarks of NE transformation is the functional inactivation of *TP53* and *RB1*, which occurs through either genomic alterations or protein downregulation, typically early in the transformation process, and may serve as a licensing condition, necessary but not sufficient for histologic transformation. Recent analyses revealed that, similar to de novo SCLC, XPO1 mRNA levels increased in pre-transformed NSCLC clinical specimens from patients carrying concomitant *TP53* and *RB1* inactivation, with double *TP53/RB1*-mutated samples showing the highest XPO1 expression relative to their double wild-type counterparts. The mechanisms contributing to the apparent increased dependence of NE tumors on nuclear transport by XPO1 have not been determined, but they likely involve the aberrant transport of transcripts controlling cell cycle regulatory and DNA damage repair pathways, on which tumor cells become increasingly dependent after the loss of *TP53* and *RB1*. In lung cancer xenograft models, XPO1 inhibition interfered with NE transformation by downregulating SOX2, a key driver of NSCLC-to-SCLC lineage plasticity, thereby prolonging the efficacy of targeted therapies and supporting XPO1 inhibition as a strategy to constrain NE reprogramming [78]. Given the clinical availability of potent and well-tolerated XPO1 inhibitors, these findings highlight a feasible approach for preventing and treating NE transformation in lung adenocarcinomas, a context where treatment options remain limited.

## 6. Predictive Biomarkers for XPO1 Inhibition in Lung Cancer

Effective implementation of XPO1 inhibitors in NSCLC requires a biomarker-guided approach, as genetic and cancer-subtype contexts strongly influence therapeutic response.

### 6.1. TP53 Status

*TP53* status has emerged as one of the most relevant predictors of sensitivity to XPO1 blockade. In *TP53*-wild-type NSCLC, selinexor promotes nuclear retention and stabilization of p53, leading to transcriptional activation of pro-apoptotic genes such as *PUMA*, *BAX*, *NOXA*, and *CDKN1A* (p21), thereby restoring chemosensitivity and potentiating the efficacy of standard cytotoxic therapies. Clinically, this mechanism translated into a median progression-free survival (PFS) of 7.4 months in *TP53* wild-type versus 1.8 months in *TP53*-mutant tumors in a Phase I/II study of selinexor combined with docetaxel [34]. These findings indicate that *TP53* wild-type status may serve as a key biomarker for patient selection in XPO1-directed clinical trials.

Mechanistically, inactivation of *TP53* has been shown to increase XPO1 promoter activity, suggesting that p53 binding can repress XPO1 transcription. Preclinical studies further demonstrate that XPO1 inhibition potentiates p53 function in wild-type contexts, promoting apoptosis and cell-cycle arrest. Conversely, in *TP53*-mutant NSCLC, the therapeutic benefit of XPO1 inhibition is diminished, as nuclear retention of mutant p53 may paradoxically support tumor progression by competing with other tumor suppressors and regulators. Consistent with this model, *TP53*-mutant cell lines display resistance to selinexor, while exploratory analyses in different cancers, such as endometrial carcinoma, have observed efficacy only in *TP53* wild-type tumors. These observations underscore the necessity of biomarker-driven patient stratification when considering XPO1-targeted therapy (NCT03555422) [33].

Partially against this idea, other studies have shown that selinexor can also exert p53-independent effects. Across NSCLC lines, including *TP53*-mutant cells, selinexor induces cell-cycle arrest and inhibits proliferation. When combined with cisplatin, it enhances cytotoxicity by downregulating NF-κB, a key regulator of survival pathways. In KRAS-mutant NSCLC, XPO1 inhibition produces strong synthetic lethal effects by increasing the phosphorylated inhibitory subunit of NF-κB, thereby further suppressing NF-κB activity and promoting apoptosis. These findings suggest that XPO1-targeted therapies may be beneficial even in genetically complex or p53-deficient tumors [18,59].

Collectively, these results highlight the central role of XPO1 in modulating tumor suppressor activity and suggest that nuclear export inhibition represents a multifaceted strategy to overcome both p53-dependent and -independent mechanisms of NSCLC progression.

### 6.2. HDAC7/MYC Subtype in SCLC

SCLC is a highly aggressive malignancy with limited treatment options. Despite advances in immunotherapy, response rates remain low, and the efficacy of current molecular subtyping is insufficient to predict therapeutic outcomes.

In 2022, Quintanal-Villalonga et al. reported that XPO1 was expressed in SCLC cell lines at levels similar to those observed in relevant hematologic malignancies, and this expression was confirmed in SCLC patient specimens [79]. In vivo, the XPO1 inhibitor, combined with chemotherapy, demonstrated stronger anti-tumor effects and prolonged survival in SCLC compared with chemotherapy alone, suggesting that the combination of selinexor with either cisplatin or irinotecan may have beneficial effects and an acceptable toxicity profile. Remarkably, the efficacy of this combination was validated regardless of the SCLC molecular subtypes, as well as in both treatment-naive and chemotherapy-resistant PDX models. However, sensitivity to selinexor monotherapy showed high variability among the PDX models treated, suggesting that some SCLC tumors may have more intrinsic dependence on XPO1 function than others. This discovery has led to the identification of a novel SCLC subtype, high histone deacetylase 7-positive (HDAC7+) SCLC, characterized by elevated HDAC7, Myc, and XPO1 expression [80]. The authors uncovered a complex positive feedback loop where they mutually regulate each other, thereby promoting cell proliferation, colony formation, and tumor progression. In preclinical models, tumors with elevated HDAC7/Myc expression exhibit markedly enhanced sensitivity [72] to XPO1 inhibitors, compared to non-HDAC7+ SCLC, proposing HDAC as an essential biomarker for stratifying SCLC patients for XPO1-targeted therapy.

## 7. Mechanisms of Resistance to XPO1 Inhibition

Despite its therapeutic promise, adaptive resistance to XPO1 inhibition is increasingly recognized, with multiple signaling routes contributing to drug tolerance and relapse (Figure 3). One critical mechanism involves activation of Yes-associated protein 1 (YAP1) by focal adhesion kinase (FAK), a downstream effector of the Hippo pathway, which promotes cell survival and proliferation even in the presence of XPO1 blockade [81]. YAP1 drives the expression of oncogenic proteins that must be transported into the cytoplasm to function, a process facilitated by RAS-mediated phosphorylation and subsequent activation of XPO1. This nuclear–cytoplasmic shuttling is especially relevant for anti-apoptotic proteins, which, instead of remaining inactive in the nucleus, are exported into the cytoplasm, where they effectively suppress apoptosis [82]. In resistant cells, FAK signature gene expression is markedly increased, reinforcing YAP nuclear entry and direct phosphorylation on Tyr357, thereby boosting its transcriptional activity. Importantly, FAK and its associated kinase ILK are putative XPO1 cargoes, further linking nuclear export to resistance mechanisms [78]. The mislocalization of YAP and Hippo pathway effector proteins is implicated in the recurrence of cholangiocarcinoma [83].

## 8. Conclusions

XPO1 is a central regulator of nucleocytoplasmic trafficking, controlling the localization and function of key tumor suppressors, oncogenes, and RNAs. Its dysregulation contributes to proliferation, survival, therapy resistance, and immune evasion across lung cancer subtypes. Pharmacologic inhibition of XPO1 with selective inhibitors such as selinexor disrupts multiple oncogenic pathways, restores tumor suppressor activity, modulates the tumor microenvironment, and enhances responses to standard therapies, including targeted agents and adoptive immunotherapies. Importantly, XPO1 inhibition shows different efficacy in *TP53* wild-type and mutant contexts, indicating that biomarker-driven stratification of patients may optimize therapeutic benefit. Preclinical and early clinical studies suggest that combining XPO1 inhibitors with chemotherapy, targeted therapies, or immunotherapies could overcome drug resistance, delay neuroendocrine transformation, and improve patient outcomes. These findings establish XPO1 as a promising therapeutic target in lung cancer, warranting continued investigation to refine combination strategies and identify predictive biomarkers for personalized therapy.

## Figures and Tables

**Figure 1 cells-14-01991-f001:**
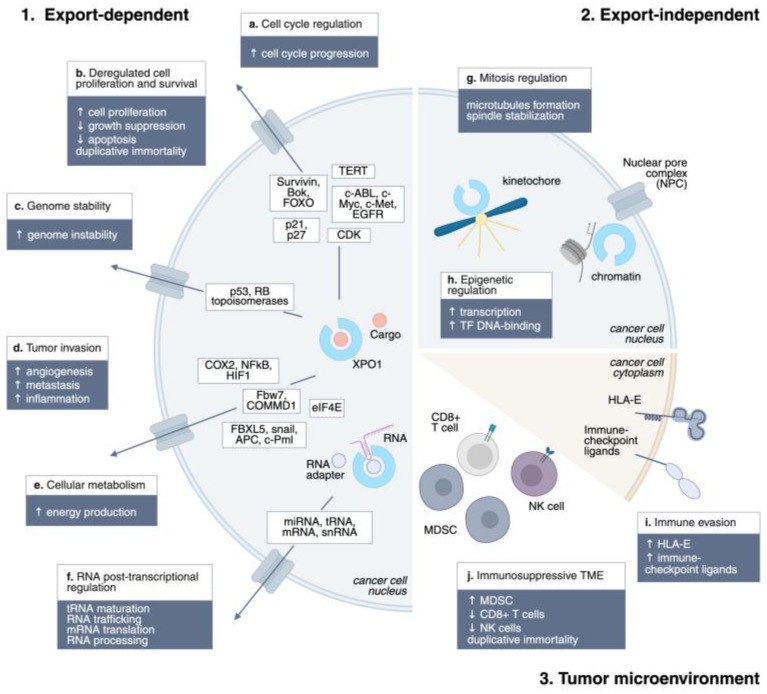
XPO1 mechanisms of action in cancer. Schematic representation of the main XPO1 mechanisms of action in cancer: 1. Export-dependent; 2. Export-independent; 3. Tumor microenvironment. ↑, upregulation; ↓, downregulation.

**Figure 2 cells-14-01991-f002:**
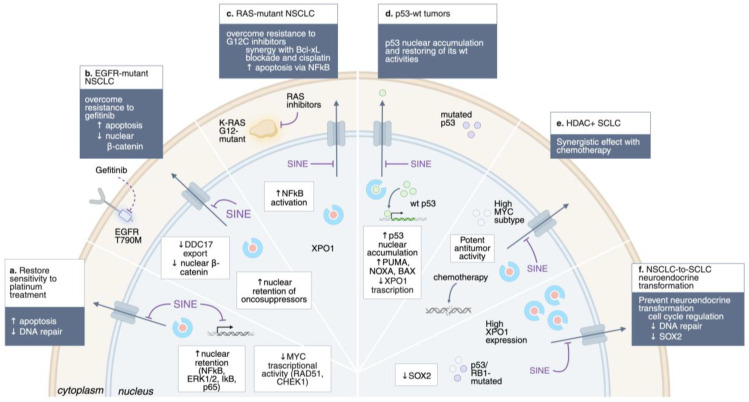
Overcoming drug resistance by XPO1 inhibition. Schematic representation of the main approach to overcome drug resistance by XPO1 inhibition in lung cancer. ↑, upregulation; ↓, downregulation.

**Figure 3 cells-14-01991-f003:**
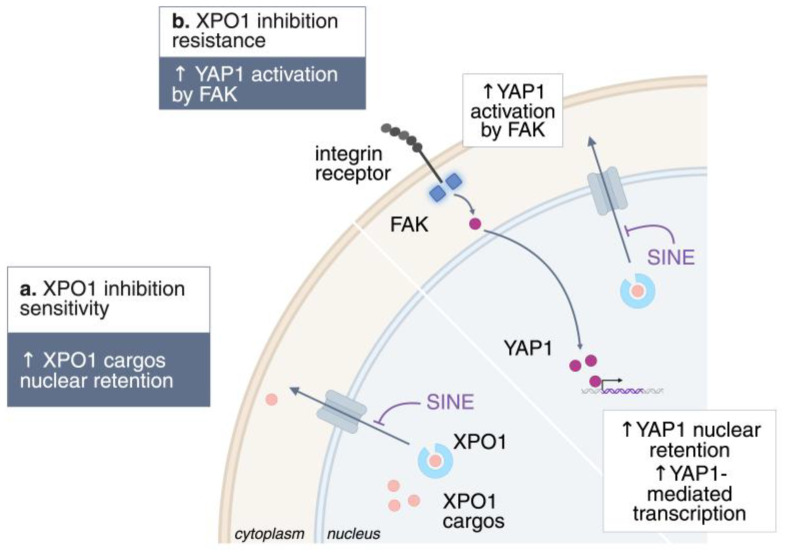
Mechanisms of resistance to XPO1 inhibition. Schematic representation of the known mechanisms of resistance to XPO1 inhibition. ↑, upregulation; ↓, downregulation.

## Data Availability

No original data were generated for this manuscript.

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
