# Peer review of "Exportin 1 as a Therapeutic Target to Overcome Drug Resistance in Lung Cancer"

_cells, 2025, doi:10.3390/cells14241991_

Round 1
Reviewer 1 Report
Comments and Suggestions for Authors
This review summarizes the current knowledge about XPO1 function and role in cancer. The manuscript provides balanced and systematic overview of XPO1 mechanism of action, molecular partners and functional consequences of dysregulated XPO1 function. Overall, the manuscript is of a high quality and may be a useful source for a broad audience.
There is one point that needs attention:
1. Figures must be edited (redrawn) - the labels are unreadable and molecular targets to small. A lot of space is occupied by empty background
Author Response
Reviewer1: This review summarizes the current knowledge about XPO1 function and role in cancer. The manuscript provides balanced and systematic overview of XPO1 mechanism of action, molecular partners and functional consequences of dysregulated XPO1 function. Overall, the manuscript is of a high quality and may be a useful source for a broad audience.
There is one point that needs attention:
1. Figures must be edited (redrawn) - the labels are unreadable and molecular targets to small. A lot of space is occupied by empty background
Reply 1: We thank the reviewer for the comment. We revised the manuscript figures accordingly to the suggestions.
Reviewer 2 Report
Comments and Suggestions for Authors
The manuscript, titled "Exportin 1 as a therapeutic target for overcom drug resistance in lung cancer," discusses the literature on the role of XPO1 in oncogenesis, maintaining the malignant phenotype, and acquiring drug resistance in lung cancer. Due to the frequent overexpression of XPO1 in NSCLC, key regulators such as TP53, FOXO, and RB1 are mislocated. These data indicate that XPO1 can be considered a potential oncogene. Inhibiting its activity could expand anticancer therapy. The manuscript is very well written and easy to read. The text is well-organized, so after reading it, one can gain a complete understanding of the topic. The only thing I would improve is the size of the figures. The text in the figures is very small, making it almost impossible to read and analyze them clearly.
Author Response
Reviewer 2: The manuscript, titled "Exportin 1 as a therapeutic target for overcom drug resistance in lung cancer," discusses the literature on the role of XPO1 in oncogenesis, maintaining the malignant phenotype, and acquiring drug resistance in lung cancer. Due to the frequent overexpression of XPO1 in NSCLC, key regulators such as TP53, FOXO, and RB1 are mislocated. These data indicate that XPO1 can be considered a potential oncogene. Inhibiting its activity could expand anticancer therapy. The manuscript is very well written and easy to read. The text is well-organized, so after reading it, one can gain a complete understanding of the topic. The only thing I would improve is the size of the figures. The text in the figures is very small, making it almost impossible to read and analyze them clearly.
Reply 2: We thank the reviewer for the comment. We revised the manuscript figures accordingly to the suggestions.
Reviewer 3 Report
Comments and Suggestions for Authors
1.This research focused on Exportin 1 as a therapeutic target to overcome drug resistance in lung cancer, after check in pubmed, not so many references about this topic(PMID: 30613477), this was mean this manuscript was with some Innovation.
2.Whole manuscript contained so much data and contents, but I think some places can be more perfect.
- This review article discusses the immune targeted therapy of lung cancer and the mechanism of drug resistance, and systematically expounds that Exportin 1 plays an important role, very interesting and with great scientific significance.
- XPO1 structure if can add a Figure maybe much better, some drug tagged XPO1 or clinal trial if add a Table maybe much better.
- The background of the Figures is a little dark. It may be better if it can be adjusted.
- What is the relationship between this gene XPO1 and common gene mutations such as EGFR、ALK、KRAS、ROS1in lung cancer.
Author Response
Reviewer 3:
1.This research focused on Exportin 1 as a therapeutic target to overcome drug resistance in lung cancer, after check in pubmed, not so many references about this topic(PMID: 30613477), this was mean this manuscript was with some Innovation.
2.Whole manuscript contained so much data and contents, but I think some places can be more perfect.
- This review article discusses the immune targeted therapy of lung cancer and the mechanism of drug resistance, and systematically expounds that Exportin 1 plays an important role, very interesting and with great scientific significance.
Reply 3: We thank the reviewer for their comments and positive feedback on the manuscript's significance.
- XPO1 structure if can add a Figure maybe much better, some drug tagged XPO1 or clinal trial if add a Table maybe much better.
We thank the reviewer for the suggestion; however, several reviews have already summarized and discussed either the drug-target complex at a structural level or the clinical trials involving SINE compounds (see Balasubramanian et al Leukemia 2022; Metzger et al Blood Neoplasia 2024). To clarify some mechanistic features of SINE pharmacodynamics, we included language in the manuscript and new references. We presented recent evidence: “These orally administered compounds bind to Cys528 on XPO1 in a slow, reversible manner, offering a better safety profile [5,26,27]. In addition, unlike LMB, many SINEs not only inhibit XPO1 function but also promote its proteasomal degradation, leading to a progressive decrease in protein levels. This reduction results from proteasome-dependent degradation rather than reduced transcription, indicating a dual mechanism whereby SINEs exert both covalent inhibition of export activity and active elimination of the export receptor itself [28].”
- The background of the Figures is a little dark. It may be better if it can be adjusted.
We thank the reviewer for the comment. We revised the manuscript figures accordingly to the suggestions. Moreover, we improved the figures' readability by increasing the font sizes of the free text and labels.
- What is the relationship between this gene XPO1 and common gene mutations such as EGFR、ALK、KRAS、ROS1in lung cancer.
We thank the reviewer for this insightful comment. We agree that defining the relationship between XPO1 and major oncogenic drivers is essential for understanding its therapeutic relevance in NSCLC. As noted in the manuscript, we have already addressed these interactions in detail in the sections discussing TKI resistance mechanisms and lineage plasticity. Specifically, the manuscript describes:
- EGFR-mutant NSCLC: XPO1-dependent nuclear export contributes to TKI resistance (including T790M and bypass signaling), and XPO1 inhibition resensitizes resistant cells by restoring nuclear localization of tumor suppressors and modulating DDX17-mediated β-catenin activity.
- KRAS-mutant NSCLC: XPO1 supports survival pathways that limit KRAS G12C inhibitor efficacy; selinexor restores sensitivity and synergizes with cytotoxic agents in vitro and in PDX models.
- ALK-rearranged NSCLC: Although less extensively studied, recent preclinical data show strong synergy between SINE compounds and ALK-TKIs, particularly in resistant models.
- ROS1-rearranged NSCLC: ROS1-driven tumors can undergo neuroendocrine transformation, a context in which XPO1 is upregulated, and XPO1 inhibition interferes with lineage plasticity.
To improve clarity, we have added explicit linking sentences in the revised version (in red in the revised manuscript body; Paragraph: Non-small cell lung cancer, subparagraph: Resistance to targeted therapy) to more clearly emphasize how XPO1 intersects with EGFR-, KRAS-, ALK-, and ROS1-driven tumor biology and therapy resistance. We hope this clarification adequately addresses the reviewer’s comment.